# Milk Quality Parameters of Raw Milk in Ecuador between 2010 and 2020: A Systematic Literature Review and Meta-Analysis

**DOI:** 10.3390/foods11213351

**Published:** 2022-10-25

**Authors:** Byron Puga-Torres, Eduardo Aragón Vásquez, Lenin Ron, Vanessa Álvarez, Salomé Bonilla, Aldair Guzmán, Diego Lara, David De la Torre

**Affiliations:** Facultad de Medicina Veterinaria y Zootecnia, Universidad Central del Ecuador, Quito 170521, Ecuador

**Keywords:** raw milk, milk quality parameters, Ecuador, meta-analysis

## Abstract

With the objective of evaluating the quality parameters of raw milk in Ecuador between 2010 and 2020, a systematic review and meta-analysis of 73 studies on raw milk produced in different regions of Ecuador was performed. Under the random effects model, effect size and heterogeneity were determined vs. climatic region both among analyses and studies, with Cochran’s Q, I^2^ and Tau (π) statistics. For all the variables, it was observed that there was great heterogeneity (I^2^ > 90%) among the studies; additionally, it was found that climatic region had an influence only among the variables arsenic, mercury, pH and total solids, and it was greater in the coastal region than the Inter-Andean region. The mean values of the physicochemical characteristics of the milk (titratable acidity, ash, cryoscopy, fat, lactose, pH, protein, non-fat solids and total solids) in the great majority of these studies were within the range allowed by Ecuadorian regulations. As for the hygienic quality of raw milk (total bacterial count, somatic cell count and presence of reductase), although the mean values were within those determined by local legislation, it should be noted that the range established by Ecuadorian regulations is relatively much higher compared to other regulations, which possibly means that there is a high presence of bacteria and somatic cells in raw milk. Finally, the presence of several adulterants (added water) and contaminants (AFM1, antibiotics and heavy metals) was confirmed in the milk, in addition to other substances such as eprinomectin, zearalenone and ptaquilosides, whose presence can be very dangerous, because they can be hepatotoxic, immunotoxic and even carcinogenic. In conclusion, there is great variability among the studies reviewed, with the physicochemical characteristics being the most compliant with Ecuadorian legislation; the hygienic characteristics, adulterants and contaminants of raw milk require greater attention by producers and local authorities, so that they do not harm the health of consumers and the profitability of producers in Ecuador.

## 1. Introduction

Systematic reviews are a set of studies that aim to answer a research question, in which an exhaustive search of the available information (studies that answer the research question) and synthesis of the results found in such research are performed; such a procedure requires a critical, reproducible and transparent methodology [1]. Meta-analysis is a statistical tool that integrates, synthesizes and quantifies the results that have been published on a variable of interest, based on predefined, clear and reproducible criteria, which allows a reduction of the biases that are commonly present in other types of reviews. In addition, it allows researchers to obtain a measure of the effect that certain specific factors may have on the response variable in a more accurate way compared to individual studies [2].

Milk is essential for people’s nutrition due to its great contribution of nutrients and biofunctional molecules, so carrying out controls and studies that guarantee its safety is of utmost importance for public health [3]. In Ecuador, the dairy industry is one of the most important economic activities involving livestock [4], where greater emphasis has been given to the production of milk of optimal quality, from a compositional point of view, because the price of milk for the producer is based on these characteristics [5,6,7]. In 2021, Ecuadorian milk production was 5.70 million liters per day, which was 7.31% lower than in 2020. The largest milk-producing province is Pichincha, followed by Azuay, and then, Manabí, where 74.85% of the total production is sold in liquid form, 16.39% is processed on the land, 6.76% is used for calf consumption, 1.87% is used for feeding from the bucket and 0.13% is milk wasted on land [8]. The rate of fluid-milk consumption is approximately 110 L per inhabitant each year [9]. The necessity for constant improvement in Ecuador’s dairy sector and globalization demand greater efforts in order to achieve efficient productivity and competitiveness; this makes the formulation and implementation of evaluation and control strategies indispensable for the diagnosis of situations [10,11]. In Ecuador there are several independent studies that are diverse and that have not been systematically analyzed within a period of time, so the integration of information on the topic of interest has not been performed. Therefore, the general objective of the present research was to analyze the physicochemical characteristics (fat, protein, lactose, total solids, non-fat solids and ash) and hygienic characteristics (bacteria, somatic cells and reductase), as well as the presence of contaminants (antibiotics, mycotoxins, heavy metals, preservatives and neutralizers) and adulterants (added water, starches and chlorides) in raw milk produced in Ecuador, through a systematic literature review and meta-analysis, in studies conducted between 2010 and 2020.

## 2. Materials and Methods

### 2.1. Search Strategies

We searched for studies that were published in certain repositories and indexed, and undergraduate and graduate degree works, between January 2010 and December 2020, that were carried out in Ecuador. A search of various studies in university repositories and indexed journals was carried out, in conjunction with review by three researchers, from several electronic databases (Google Scholar, Pubmed, Elsevier, Dialnet and Science Direct (Journal)) and in the repositories of the Universities of Ecuador, of undergraduate and graduate research topics. A combination of keywords was used for the search (milk, raw, hygiene, bacteria, somatic cells, quality, Ecuador and cows, among others) both in Spanish and English.

The response variables were the hygienic characteristics of the milk (bacteria, somatic cells and pathogens), as well as the presence of contaminants (antibiotics, mycotoxins, heavy metals, preservatives and neutralizers) and adulterants (added water, starches, chlorides and vegetable fats). The moderating variable to consider was the region of Ecuador where the study was conducted.

### 2.2. Literature Inclusion Criteria and Data Extraction

We used those studies (scientific articles, undergraduate or graduate theses and publications) in which it was possible to detect and measure one or more parameters in raw milk, in terms of hygienic characteristics, composition, physicochemical properties, contaminants and adulterants; likewise, it was necessary that at least 10 observational units were available, with the mean, minimum and maximum values obtained. Publications where the methodology, results or conclusions were not clear; that were irrelevant (without important information); that were without statistical or quantitative contribution; that were without complete data; that duplicate studies; or that were conducted prior or subsequent to the study period were not taken into account.

### 2.3. Statistical Analysis

The data obtained for the variables of interest were tabulated in a Microsoft Excel spreadsheet. For statistical processing of the data, the Metafor MAd package of the free statistical software RStudio, version 1.2.5019 (RStudio Inc. Boston, MA, USA) was used. The statistical significance level was set at *p* < 0.05. The data were analyzed using weighted random-effects meta-analysis models for the differences among the study means and the relationships of the studies with climate region. Statistical heterogeneity was analyzed using Cochran’s Q statistic, Higgins’ inconsistency (heterogeneity index I^2^), and the Tau-squared (π^2^) statistic. This research was not based on the recommendations indicated by PRISMA, as these are observational and multivariate studies.

## 3. Results and Discussion

### 3.1. Process of Study Selection and Searched Results

The process of choosing and discarding studies is detailed in Figure 1. Based on our criteria, the studies should include information on the sample size, the author, the year, the province of study and the ranges of values. Subsequently, the data for the meta-analysis were selected from the text, appendices, tables or graphs of each selected study and recorded in a personalized way for the systematic review. Data were extracted on the authors, the year of publication, the year of study, the complete title, the type of study, the location (coast, Inter-Andean or Oriental region), the number of samples, the results of the analysis of each variable, the statistics and the availability link of the selected documents. 

### 3.2. Overall Results of the Literature

Table 1 shows the values of each variable analyzed and found in the systematic review, with respect to the studies found and the climatic regions, based on the limits allowed by the Ecuadorian Technical Standard (NTE) INEN 9 [84], which establishes the requirements that raw milk should meet in Ecuador; it also shows the mean, minimum and maximum values of these parameters, the number of studies analyzed, the total number of samples, the percentage of samples outside the allowed range and the percentage of non-compliance.

As indicated by the analyzed studies, there is high non-compliance of the raw milk analyzed between 2010 and 2020 with respect to Ecuadorian regulations, especially regarding hygienic quality, contaminants and adulterants; this constitutes a serious public health problem. For example, in the case of heavy metals, the single study that aimed to determine lead in milk in Ecuador found that 98.28% (57/58) of the samples analyzed contained levels above the maximum level permitted by Ecuadorian regulations. Similarly, according to the analysis of mercury and arsenic, it was found that 25.64% samples (20/78) exceeded the limits allowed in food. Likewise, 89.06% (114/128) of the samples analyzed exceeded the permitted limits for the variable “Ptaquiloside”, which is a toxin of the fern *Pteridium aquilinum* that can be present in milk when the cow ingests this weed. In the case of the presence of antibiotics, it was determined that 14.55% (370/2543) of the milk samples contained them.

Regarding adulteration, the presence of water addition was found in 61.46% (118/192) of samples in three different studies, while 27.76% (457/1646) of the samples had altered values of the freezing point of milk (cryoscopy) in eight different studies. In the single study on the determination of glycomacropeptide (GMP), an indicator of milk adulteration with cheese whey, it was found to be present in 37.50% (9/24) of the samples investigated.

Regarding hygienic quality, 20.38% (22,338/109,610) of the samples had total bacterial counts higher than the maximum allowed by NTE INEN 9 (1 × 10^6^ CFU/mL); if this parameter was analyzed using international standards, the percentage of non-compliance would be alarmingly higher. This is related to indirect tests for determining bacterial contamination in milk; for example, for the reductase test, 61.80% (596/932) of the samples did not comply with Ecuadorian regulations; likewise, the parameters of pH and titratable acidity showed non-compliance of 36.06% (331/918) and 12.72% (201/1580), respectively. Among the 30 studies analyzed on somatic cell count, it was determined that 0.80% (878/110,347) presented values above the maximum allowed by NTE INEN 9, which establishes a maximum limit of 700,000 (CS/mL). However, it should be clarified that the Ecuadorian upper limit is considerably higher than the international standard (up to 400,000 CS/mL).

Regarding the presence of chemical products that are intended to mask the acidity of milk, non-compliance of 14.21% (137/951) and 6.03% (61/1011) was found for the presence of neutralizing substances and peroxides, respectively. Likewise, relative density and protein stability tests showed non-compliance of 11.30% (224/1983) and 11.70% (171/1462), respectively. Regarding AFM1—belonging to the group of aflatoxins (AF), which are extremely toxic substances from the fungus *Aspergillus*, which contaminates plant foods—when ingested by dairy cows, they are converted in the liver from AFB1 to AFM1, which is eliminated through milk [85]; AFM1 is a mycotoxin classified as potentially carcinogenic to humans (Group 2B) by the International Agency for Research on Cancer—IARC [86]. In our review, only 1% (4/401) of the samples analyzed exceeded the maximum permitted limit of AFM1 allowed by NTE INEN 9. For the parameters of chlorides, colorants, starches, the drug eprinomectin and the mycotoxin zearalenone, no non-compliance was found with respect to Ecuadorian regulations.

In the case of the chemical characteristics of the milk, the parameter of non-fat solids (NFS) presented non-compliance of 12.64% (247/1954), ashes in 7.89% (34/431), lactose in 5.10% (56/1097), protein in 0.93% (1012/109 020), fat in 0.52% (564/109 428) and total solids in 0.48% (511/106 707) of the analyzed samples; therefore, the great majority of the parameters did comply with the local legislation.

### 3.3. Meta-Analysis of Variables by Sample and by Region

Table 2 shows the statistical analysis of the meta-analysis of the parameters of raw bovine milk analyzed between 2010 and 2020 in Ecuador. There were parameters that could not be meta-analyzed, either because their values were zero (0), as in the case of starch and chloride variables, or because there was only one study of that parameter, as in the case of colorants, eprinomectin, glycomacropeptide, lead and zearalenone. For the rest of the parameters, it was observed that the heterogeneity index (I^2^), in all cases was higher than 90%, which indicates very high variability among the studies and is well above the acceptable limit (40–50%) in meta-analysis. This variability is confirmed by observing the coefficient of the *H*^2^ statistic, which is quite variable.

The results shown in Table 2 suggest that the studies included in this research work should not have been meta-analyzed since they are very different from each other; however, we proceeded to perform the statistical analysis, since they are observational studies and we wish to compare these results with respect to compliance (or a lack of it) with NTE INEN 9. In the case of the statistical analysis by study, we observed that the result of the *p*-value of Q, in all cases, was less than 0.05, which confirms the existence of significant differences among them. Regarding the statistical analysis between regions (Inter-Andean vs. coast), for the variables of added water, lactose, neutralizers and ptaquilosides, we have no comparison, since the studies were only carried out in the Inter-Andean region. Regarding the variables of arsenic, mercury, pH and total solids, a *p*-value of Q < 0.05 was observed, indicating a higher presence in the coastal region. The rest of the variables presented a *p*-value of Q > 0.05, so no significant differences were found between the regions.

### 3.4. Forest Plot of the Physicochemical Variables of Milk by Sampling and by Region

When analyzing the forest plots of titratable acidity, ash, cryoscopy, fat, lactose, pH, protein, non-fat solids and total solids, it can be observed that the average effect size, represented by a rhombus, along with most of the studies and their averages, is within the range allowed by NTE INEN 9. For example, in the case of titratable acidity (Figure 2), some studies [18,36,60] present values above what is stipulated, while the study of [38] is below what is allowed. The titratable acidity is elevated when microbiological contamination occurs, while if it is decreased, it may be due to the presence of mastitis, adulteration with water or alteration by an alkalinizing [87].

In the case of ashes, the study whose mean is below what is required (Figure 3) is that of [38]. All the studies show large variability with respect to the global mean, which is evident in the forest plot (Figure 3). 

In the case of cryoscopy, the studies of [38,42,44,79] report values, on average, outside the requirements of the Ecuadorian standard (Figure 4), with the last study mentioned being the one that shows the greatest variability with respect to the mean (Figure 4). Values approaching the freezing point of 0 °C indicate the addition of water, heating, the precipitation of phosphates in the raw milk, etc. [87].

Regarding the relative density parameter, the research of [51] presents values much lower than the minimum allowed. Regarding the forest plot (Figure 5) of the climatic period, although most of the studies comply with the Ecuadorian standard, the results found by [18,24] present greater variance with respect to the global mean. Lower values indicate adulteration with water, while higher values may indicate skimming of the milk or the use of adulterating substances such as starch or salt, since these are used to balance the density of the milk after adding water [88,89].

In most studies regarding fat percentage, it is above the minimum value allowed (3%) by local legislation. However, there are studies, such as those of [24,80], that find an amount of fat below this value. In the forest plot (Figure 6) with respect to the climatic season, it can be observed that the results found by [18,63], as well as that of [24], present greater variance with respect to the global mean. The percentage of milk fat is related to the nutrition that the animal has, as well as its breed [90], and is also dependent on the duration of grazing, the time of the year, the crop capacities and the type of pasture; the voluntary skimming of milk is one of the most common adulterations in milk [87], so the decreased values may be due to this characteristic. 

In the case of lactose values, all the studies are within the range of local regulations (Figure 7).

In the hydrogen potential (pH) parameter, the study of [24] has a mean above the stipulated value, while the studies of [18,23,26,43,44,57] are below the accepted minimum (Figure 8). The pH helps to determine certain indicators of milk quality, such as conservation, since changes in this variable can modify the stability of the protein, causing unpleasant flavors in the milk [91]. When a value below the average value is observed, it may be due to the presence of colostrum or bacterial decomposition; on the contrary, when values above the normal range are found, it is an indicator of possible mastitis or other factors due to adulterants such as neutralizers [92].

When analyzing the forest plot of protein (Figure 9), it can be observed that the average effect size is 3.23%. That is, the estimated global mean, and that of the majority of the studies, is within the range allowed by NTE INEN 9. However, the results found by [24,63], present greater variance with respect to the global mean (Figure 9). 

In the case of non-fat solids, the study whose mean is below what is required (Figure 10) is that of [44], while the results of [63,79] are those that present the greatest variance with respect to the global mean and which are evidenced in the forest plot of the climatic season (Figure 10).

Likewise, for the total solids variable, the studies of [24,80] found a percentage of total solids below that required by the Ecuadorian standard (Figure 11). Regarding the climatic season, in addition to the aforementioned studies, we also find that [18] presents greater variance with respect to the global average (Figure 11). 

### 3.5. Forest Plot of Milk Hygienic Variables by Sampling and Region

Figure 12 shows the forest plot of the total bacterial count; the plot is elaborated based on a logarithmic scale. As for the results of the model, it can be observed that the average effect size of the bacterial count in graph A, represented by a rhombus, is 13.84, which is equivalent to (Exp^13.84^), that is, 1,024,791.77 CFU/mL. Therefore, the estimated global mean and several studies are within the maximum allowed by NTE INEN 9 (≤1,500,000.00 UFC/mL); there are several studies that present bacterial counts much higher than this allowed maximum, as in the case of [23,26]. 

It should be emphasized that the range established by Ecuadorian regulations is relatively higher than that of other regulations, which means that there is a high presence of bacteria in raw milk in Ecuador; this is probably related to failures in hygiene, in the disinfection of production areas and milking procedures or in materials or equipment; bacterial growth in milking equipment; contamination by dirty cow udders; the milking of cows with mastitis; inadequately cooled milk (failure to control the storage temperature); or the quality of the water used, among others [93,94]. The importance of milk containing low bacterial counts lies in the fact that certain bacteria survive thermal processes, which could affect, in some way, the taste, texture or shelf life of milk [95], since bacteria can cause the spoilage of milk, as well as diseases that affect humans [96]. Therefore, when high microbial contents of milk are found, milk procurement needs to be improved [97]; for example, with proper disinfection, cleaning, storage and transportation of raw milk, it is guaranteed to be of good quality for consumption; likewise, by performing good milk handling practices, by training both the producers and actors involved in marketing and transportation, the quality of milk can be improved [98]. The pasteurization of milk seeks to eliminate pathogenic bacteria and guarantee the health of consumers, and it has been found that this heat treatment has a minimal effect on the nutritional characteristics of milk [99].

Reductase is not a natural enzyme in milk, but its presence is constant due to bacterial contamination, so its analysis, known as the methylene blue reduction time, is an indirect test of microbial contamination in milk. In the vast majority of samples, its presence was higher than the minimum values allowed by local legislation (Figure 13).

Figure 14 shows the forest plot of the somatic cell count; the plot is elaborated based on a logarithmic scale. As for the model results, the mean effect size of the somatic cell count in plot, represented by a rhombus, is 13.10, which is equivalent to (Exp^13.10^)—ergo, 488,942.41 CS/mL—which is represented by the rhombus. There are values found by authors, such as those of [57], that are far from the global mean, but are still within the range of the Ecuadorian standard, which establishes a maximum permissible level of ≤700,000 CS/mL.

Likewise, several studies present greater variance with respect to the global average, as is the case in [63,65,70], although the averages mostly comply with local legislation; however, if the international requirements (≤200,000 CS/mL) of milk quality are taken into account, it can be noted that very few findings would be within this range.

The somatic cell count in milk indicates the hygienic sanitary quality of the mammary gland, and is also considered a health indicator since the somatic cell count increases in direct proportion to the severity of the infectious disease [100,101]. Likewise, it directly affects milk production since it decreases milk production by 2.5% for each increase of 100 thousand somatic cells from the 200 thousand that are considered normal; therefore, it could be expected that a herd with a count of 500 thousand CS/mL would have a 7.5% decrease in production due to subclinical mastitis [102].

### 3.6. Forest Plot of Adulterant and Contaminant Variables in Raw Milk by Sampling and by Region

Regarding Aflatoxin M1, the mean is 0.04 μg/kg and its graphic representation (rhombus) is on the right side of the unit value, which means that in the raw milk analyzed in these studies, the mycotoxin is present in different quantities, most of which do not exceed the 0.5 μg/kg allowed by the NTE INEN 9 standard; however, if compared with the European standard (MLR = 0.05 μg/kg), two studies conducted by the authors of [81] in the province of Manabí and Pichincha, exceeded this limit; they also observe that climatic region is a factor related to the presence of contaminants in raw milk, which are numerically higher in the coastal region, but not statistically significant (Figure 15). AFM1 is the only mycotoxin with maximum limits allowed in milk [103], since it can be hepatotoxic and carcinogenic [104], and it is not destroyed by the pasteurization or technification of dairy products [105].

In the case of studies on the presence of added water, it was found that in the same studies, added water was present, which is illegal according to Ecuadorian legislation (Figure 16) and is carried out with the intention of increasing the volume of milk.

One of the major concerns is the high presence of antibiotic residues in the studies analyzed. In Figure 17, the forest plot of antibiotics in raw milk, where the presence of this contaminant was found, shows great variation among the studies, with between 10% and 72% positive samples in different provinces of Ecuador. 

These antimicrobials are considered one of the highest-risk contaminants, as they are widely used in cattle for the control of various diseases, and are also used in sub-therapeutic doses that are added to feed to act as growth promoters [106]. Despite being an important tool to combat diseases, excessive use and misuse can induce residues in milk when withdrawal times are not respected [107]. In this way, they become a potential risk, causing serious problems for their consumers, such as hypersensitivity, allergic reactions, bacterial resistance. In addition to this, in industrialization, it affects cheese and yogurt production, directly inhibiting the bacterial fermentation process [108,109]. These data show that control measures are inefficient in the dairy industry, so their permanent control is essential for all those involved in the dairy chain [3].

For these reasons, food control and safety agencies establish standards and monitoring programs for the maximum permissible limit of antibiotic residues in raw milk. Heat treatment plays an important role in preventing the development of antimicrobial resistance. Although antibiotic molecules are not completely degraded, this process is efficient in destroying 99.99% of bacteria that may contain resistance genes, thus preventing the multiplication of these types of bacteria; however, it is not known exactly whether the resistance genes contained in the bacteria are viable, even after pasteurization, and theories are currently being investigated [110].

In the case of neutralizing substances, most of the samples did not show the presence of these substances in the milk analyzed, but in some investigations, the use of these substances was evident, as in the results obtained in [31,32,43], as shown in Figure 18. These chemicals are used to mask the acidity of raw milk, and generally have serious consequences for public health in high doses as they are able to cause the development of kidney stones, or become deposited in body fluids and soft tissues [111,112]. 

Regarding the presence of hydrogen peroxide (which is not allowed by Ecuadorian legislation), there is wide variability among the studies; for example, in the research of [31,32,82], its levels are farthest from the zero point or the line of incidence, so they disagree with the majority of studies (Figure 19). In Ecuador, the use of hydrogen peroxide is prohibited because it is used to mask the acidity of milk [111,112]; however, there are studies that indicate the benefits of using it to preserve milk for up to 8 h at room temperature without losing its organoleptic characteristics, even in the Ecuadorian tropics, where ambient temperatures can be above 25 °C [113].

Regarding the fern toxin, called ptaquiloside, studies were found that demonstrate the presence of this variable in the samples analyzed in Ecuador (Figure 20). Cattle that ingest ferns can develop several diseases such as hemorrhagic problems, hematuria and even carcinomas, which are the carcinogenic problems that can affect humans the most [114].

Figure 21 shows the studies carried out on heavy metals in Ecuador (mercury, arsenic and lead). In the case of mercury and arsenic, neither the Ecuadorian regulations nor the Codex Alimentarius indicate maximum permissible values for raw milk, so the results were interpreted in relation to drinking water (0.001 mg/kg). The mean of these studies crosses the value of the plot unit, which means that the results do not present a relevant significance value to determine the presence of these heavy metals; this does not happen with lead, for which the majority of values are above the maximum stipulated by local and international legislation. Regarding climatic region, the highest presence of arsenic and mercury was determined in the coastal region, while for lead, there is only one study in the Sierra region. The presence of these heavy metals has a natural and anthropogenic origin. Naturally, it is documented that in localities where there are volcanic eruptions, the level of lead rises in the environment. Anthropogenic activities such as mining and refining remove high levels of heavy metals in the environment [115]. Heavy metals have genotoxic, nephrotoxic and carcinogenic properties, and also cause severe oxidative stress [116].

Based on the findings of this study, the presence of adulterants and contaminants in raw milk between 2010 and 2020 in Ecuador is evident. The findings are of great concern for producers, consumers and regulatory agencies, since the averages of the contaminants analyzed in this systematic review were: AFM1 (0.04 μg/kg), antibiotics (0.09 μg/L), lead (0.208 mg/kg), arsenic (0.01 mg/kg) and mercury (0.01 mg/kg). In addition to these substances, it should be mentioned that there were studies of public health relevance that reported the presence of several types of contaminants in raw milk, such as eprinomectin, zearalenone and ptaquilosides. Another issue is that thermal treatments are not effective at disintegrating these types of contaminants due to their thermal stability [117].

## 4. Conclusions

The systematic review and meta-analysis of 73 studies of the milk quality parameters of raw milk produced in different regions of Ecuador, between 2010 and 2020, indicates that there is great variability among them (I^2^ > 90%) with respect to the different variables analyzed; we found better compliance with the Ecuadorian regulations in the physicochemical parameters, especially the composition parameters such as fat (mean: 3.69%), protein (mean: 3.2%), lactose (mean: 4.78%), ash (mean: 0.6725%), non-fat solids (mean: 8.66%) and total solids (mean: 12.24%). Regarding hygienic quality (total bacterial count and somatic cell count), the local regulations are very lenient compared to other regulations, which means that there is a high presence of bacteria in Ecuadorian raw milk (mean: 6,878,541.1 UFC/mL); this is probably related to hygiene failures in milking and in the storage and transportation of milk, and the high somatic cell count (mean: 695,736.1 CS/mL). Likewise, adulterants and contaminants in raw milk have been determined in several studies, which is a cause for concern (for example lead (mean: 0.208 mg/kg), AFM1 (mean: 0.421 µg/kg), antibiotics (14.55%), arsenic (mean: 0.005 mg/L) and mercury (mean: 0.00009 mg/L)). It is necessary to take corrective actions, through training in producers, to improve the quality of milk produced in Ecuador, which will benefit the public health of consumers and the profitability of livestock farms.

## Figures and Tables

**Figure 1 foods-11-03351-f001:**
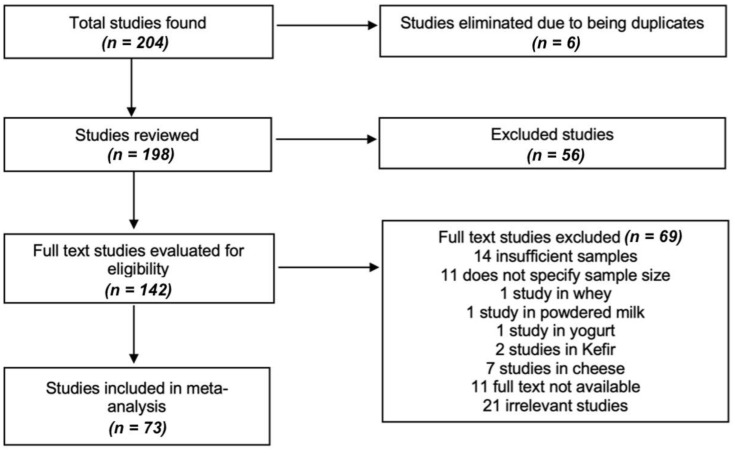
Flow chart summarizing the process of study selection. We identified 204 studies from which 198 unique references were retrieved, where we evaluated the full text of 142 studies, of which 69 studies were excluded and only 73 studies [12,13,14,15,16,17,18,19,20,21,22,23,24,25,26,27,28,29,30,31,32,33,34,35,36,37,38,39,40,41,42,43,44,45,46,47,48,49,50,51,52,53,54,55,56,57,58,59,60,61,62,63,64,65,66,67,68,69,70,71,72,73,74,75,76,77,78,79,80,81,82,83] were meta-analyzed (view: Appendix A); no additional studies were identified after updating the literature search.

**Figure 2 foods-11-03351-f002:**
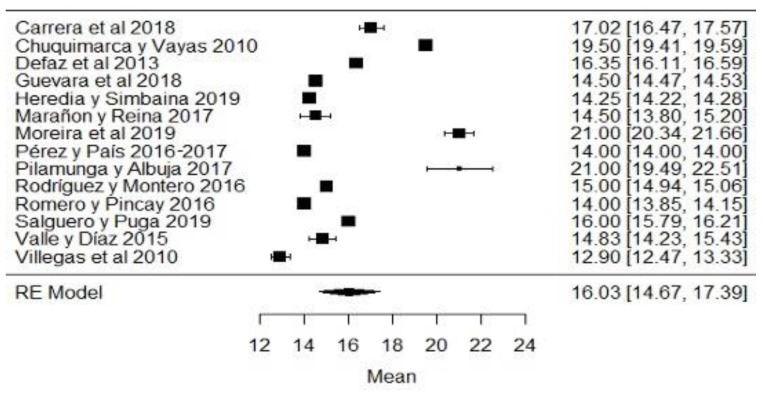
Forest plot of titratable acidity by study [16,18,21,23,24,26,35,36,38,39,44,60,61,79].

**Figure 3 foods-11-03351-f003:**
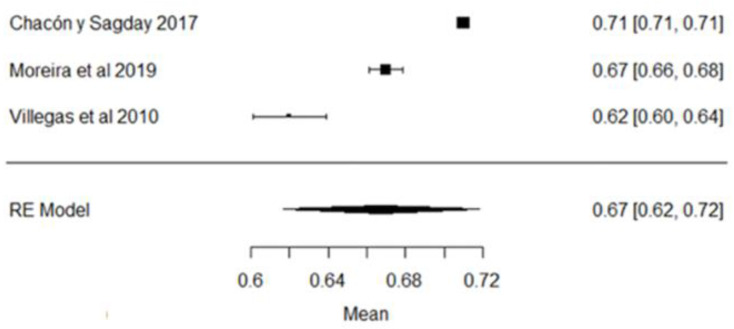
Ash forest plot [18,38,42].

**Figure 4 foods-11-03351-f004:**
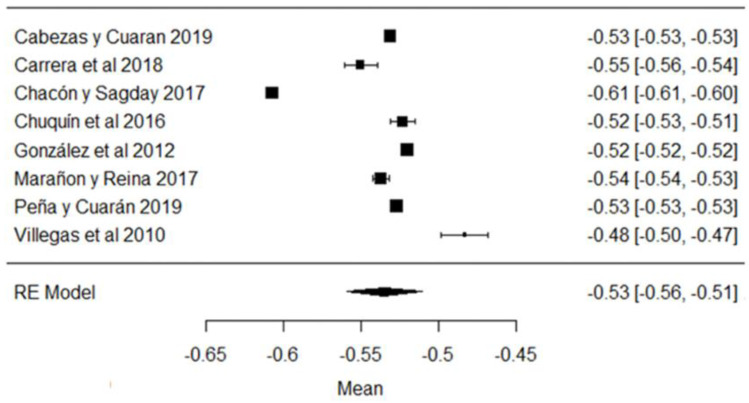
Cryoscopy forest plot [14,33,38,42,44,53,69,79].

**Figure 5 foods-11-03351-f005:**
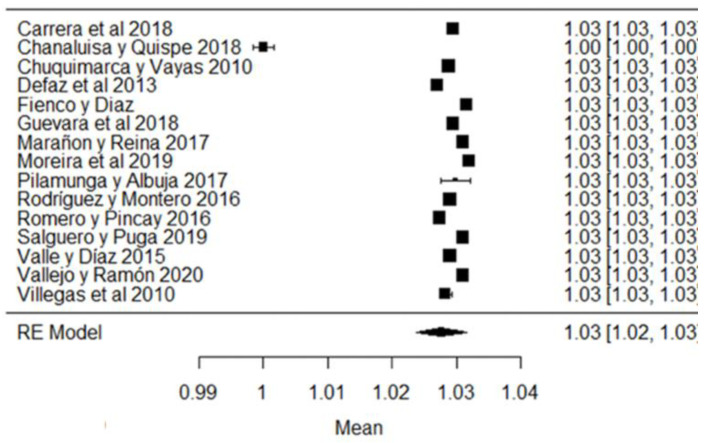
Forest plot of relative density [18,21,23,24,26,36,38,39,44,51,57,60,61,66,79].

**Figure 6 foods-11-03351-f006:**
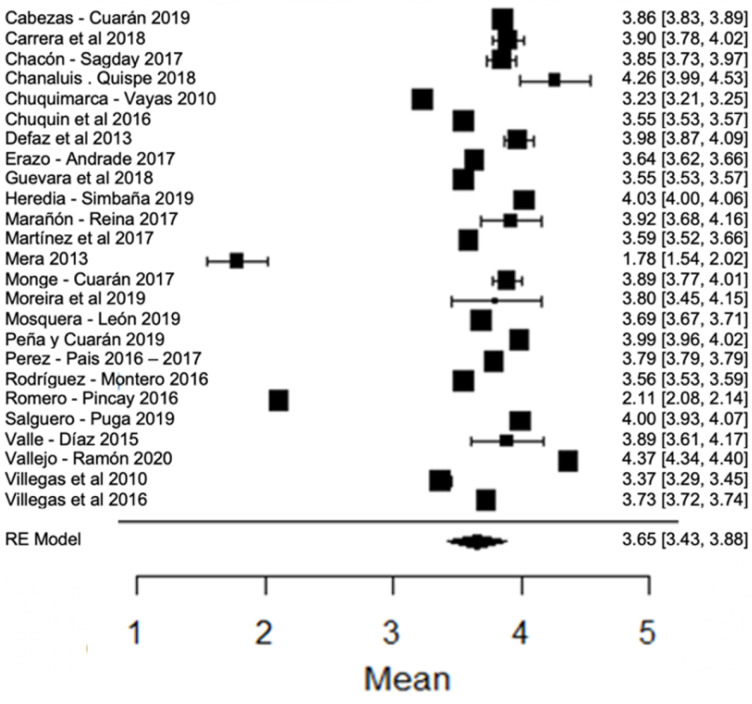
Forest plot of fat [14,16,17,18,21,23,24,26,28,33,35,38,39,42,44,51,53,57,60,61,63,76,77,79,80].

**Figure 7 foods-11-03351-f007:**
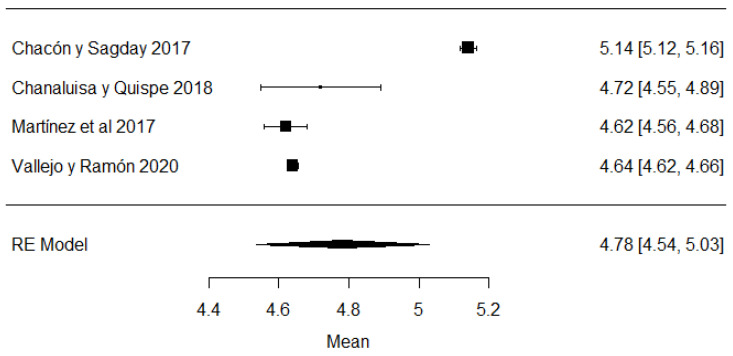
Lactose forest plot [42,51,57,77].

**Figure 8 foods-11-03351-f008:**
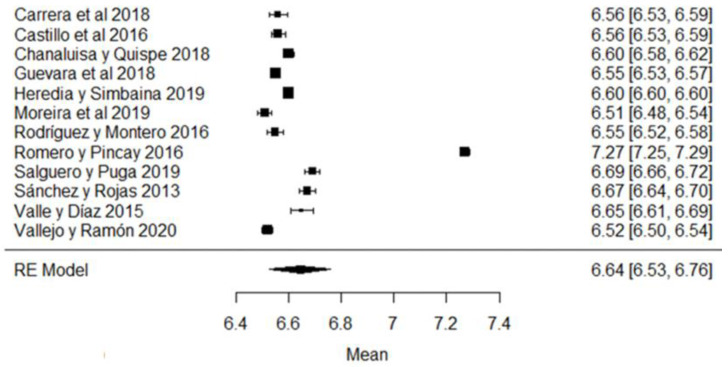
Forest plot of pH [16,18,21,23,24,25,26,39,43,44,51,57].

**Figure 9 foods-11-03351-f009:**
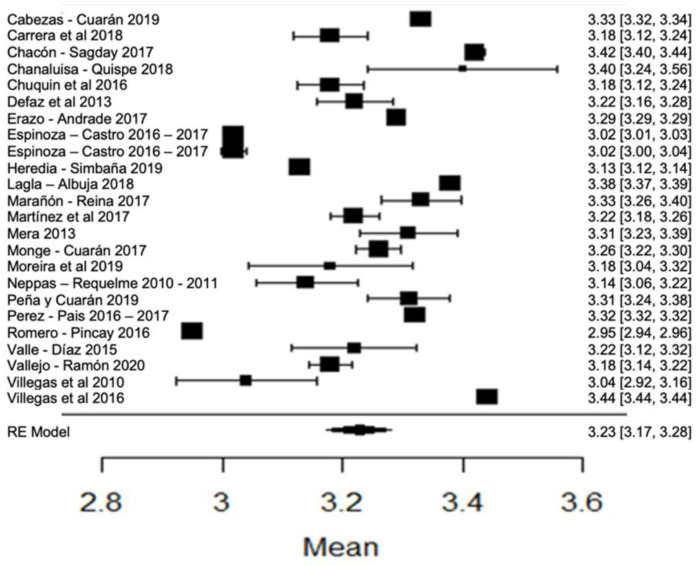
Forest plot of protein [14,16,17,18,21,23,24,26,28,30,33,35,38,39,42,44,51,53,57,60,61,63,64,76,77,79,80].

**Figure 10 foods-11-03351-f010:**
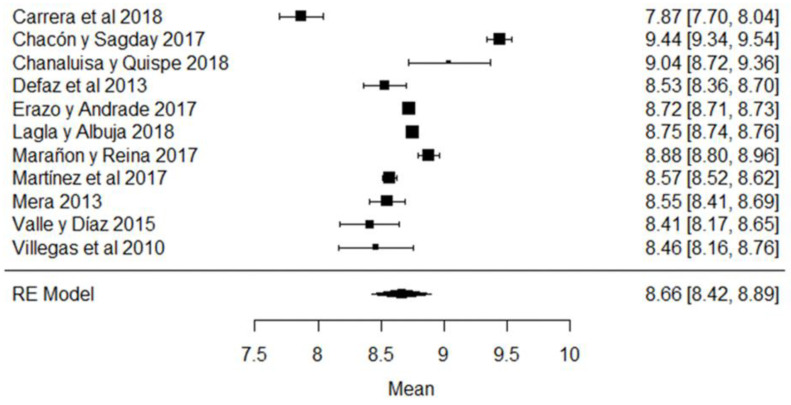
Forest plot of non-fat solids [38,39,42,44,51,61,63,73,77,79,80].

**Figure 11 foods-11-03351-f011:**
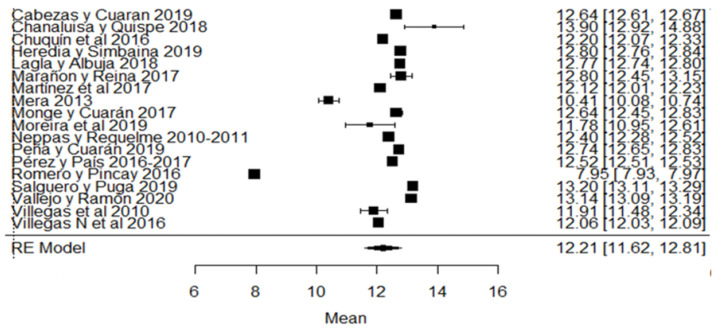
Forest plot of total solids [14,16,17,18,21,24,30,33,35,38,51,53,57,73,76,77,79,80].

**Figure 12 foods-11-03351-f012:**
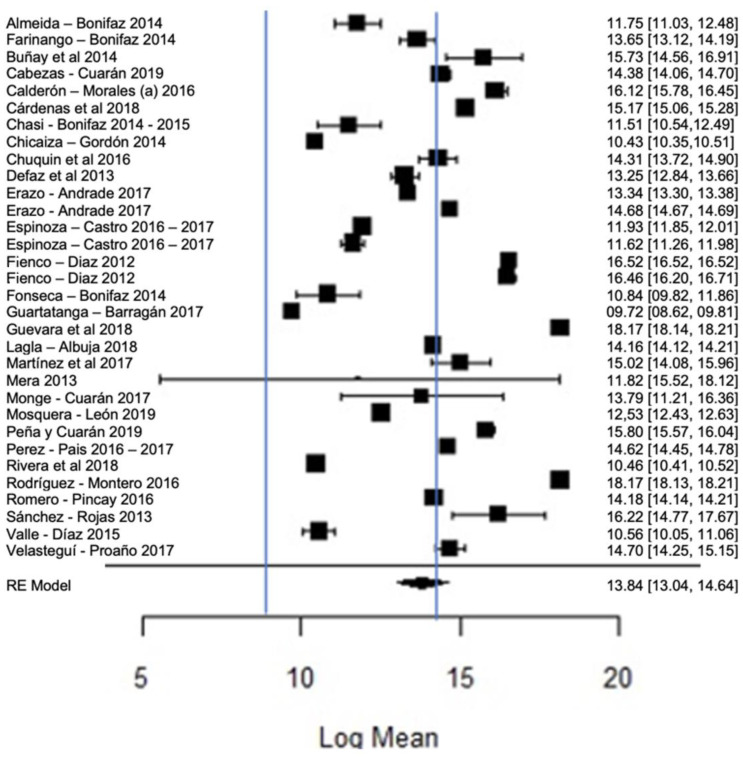
Forest plot of the bacterial count [14,23,24,25,26,27,28,29,33,35,39,47,52,53,54,56,58,61,63,64,66,67,68,71,73,76,77,80].

**Figure 13 foods-11-03351-f013:**
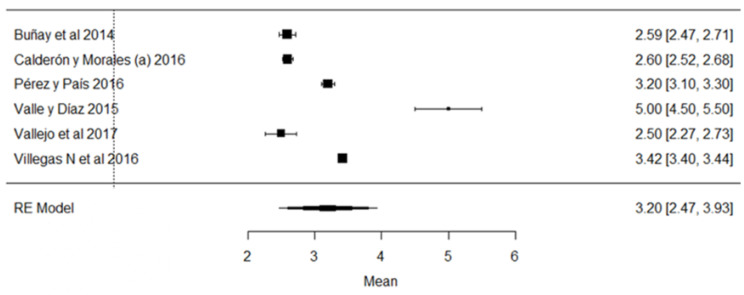
Forest plot of reductase presence [19,35,38,39,52,54].

**Figure 14 foods-11-03351-f014:**
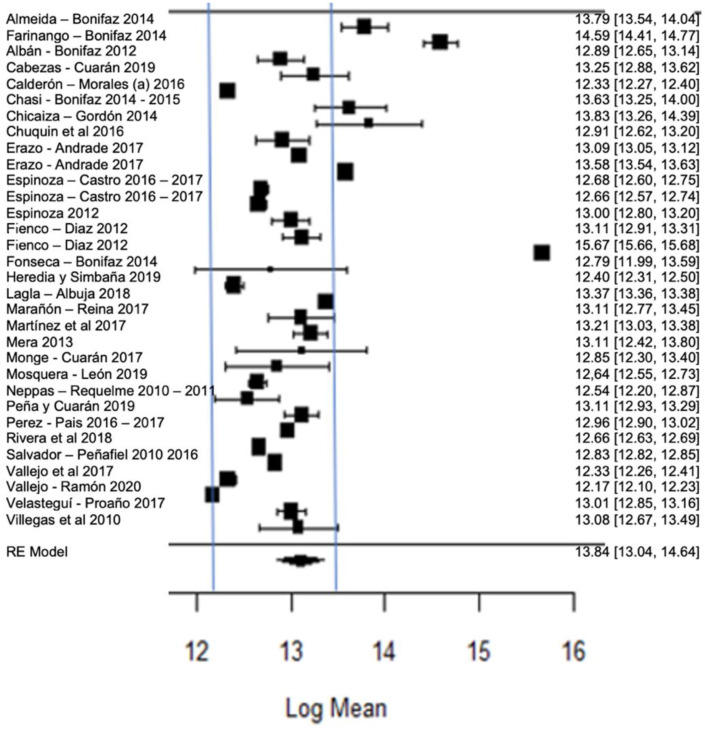
Forest plot of somatic cell counts [14,23,24,25,26,27,28,29,33,35,38,39,47,52,53,54,56,58,61,63,64,66,67,68,71,73,76,77,80].

**Figure 15 foods-11-03351-f015:**
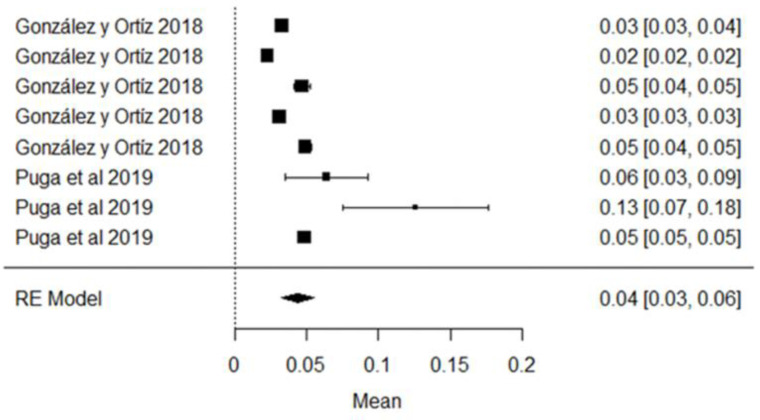
AFM1 forest plot [40,81].

**Figure 16 foods-11-03351-f016:**
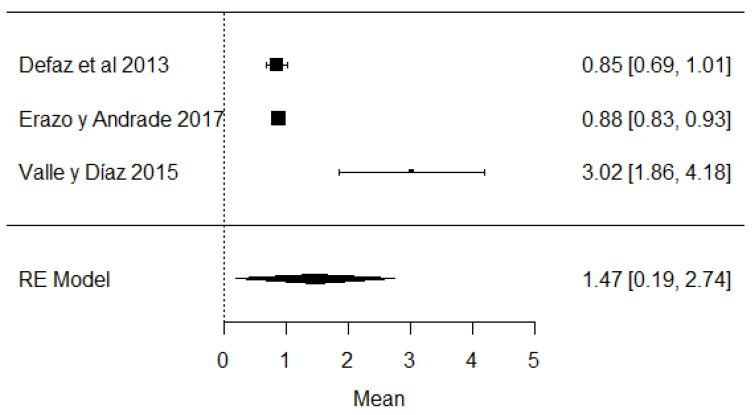
Forest plot of added water [39,61,63].

**Figure 17 foods-11-03351-f017:**
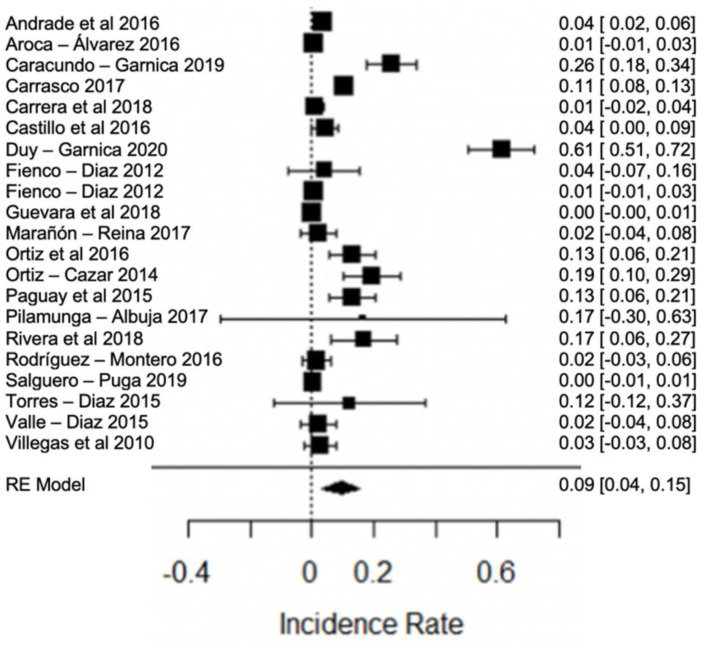
Forest plot of antibiotic [21,23,26,27,31,32,36,38,39,43,44,46,48,49,50,62,66,79,82].

**Figure 18 foods-11-03351-f018:**
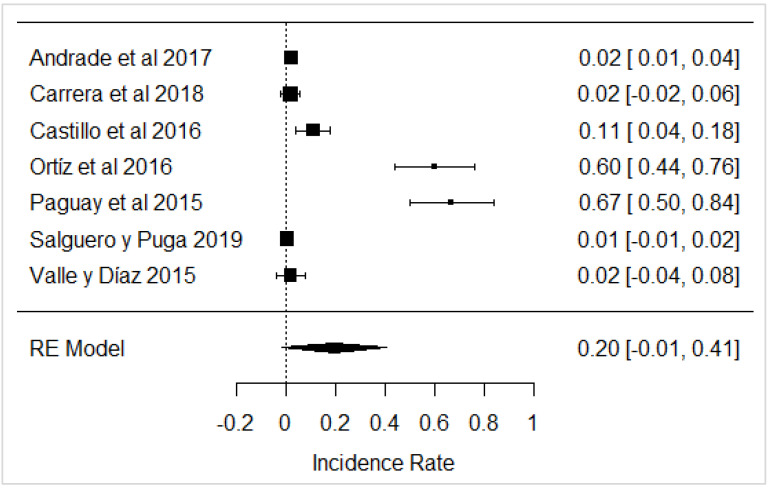
Forest plot of neutralizing agents [21,31,32,39,43,44,82].

**Figure 19 foods-11-03351-f019:**
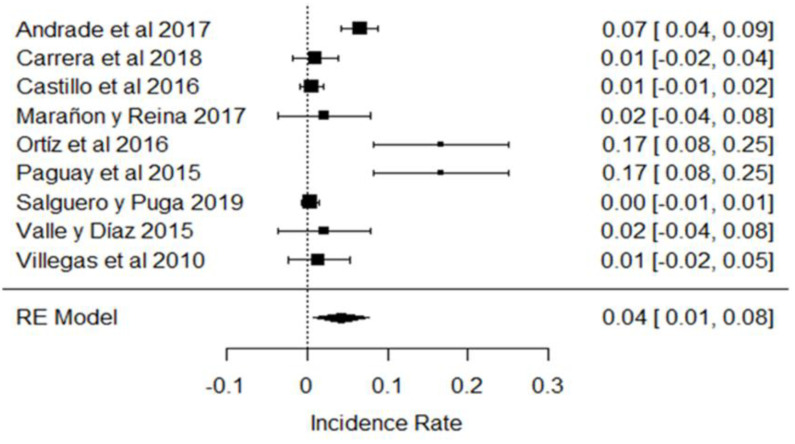
Forest plot of the presence of peroxides [21,31,32,38,39,43,44,79,82].

**Figure 20 foods-11-03351-f020:**
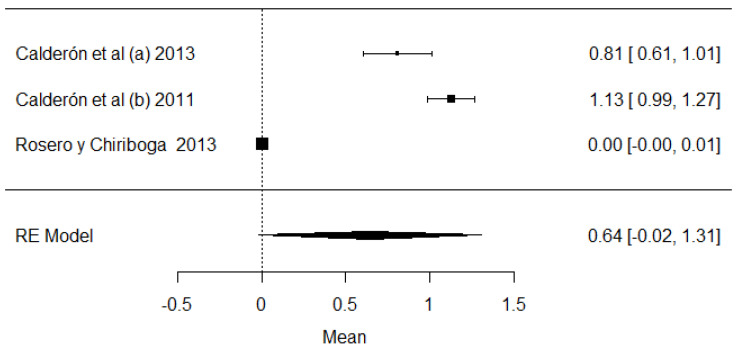
Forest plot of ptaquilosides [15,22,70].

**Figure 21 foods-11-03351-f021:**
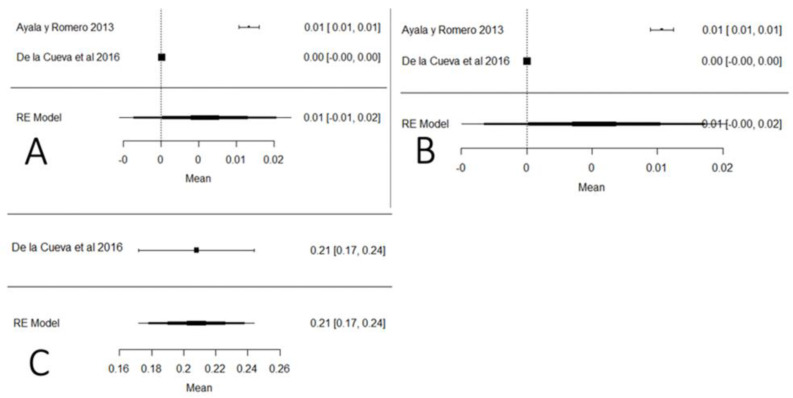
Forest plot for the presence of heavy metals: (**A**) mercury; (**B**) arsenic; (**C**) lead [12,83].

**Table 1 foods-11-03351-t001:** Allowable limits, measures of central tendency, studies analyzed and % compliance of the variables used in the study.

Variable	Type (Total/Region)	Permitted Limit NTE INEN 9	Mean	Minimum Value	Maximum Value	Number of Studies Analyzed (*n* = 74)	Total Samples	Number of Samples Out of Range	Percentage of Samples Out of Range
Titulable Acidity (°D)	Total	13–17	15.84	10	25	19	1580	201	12.72%
	Inter-Andean		15.48	10	25	15	1415	101	7.14%
	Coast		17	11	22	4	165	10	6.06%
AFM1 (µg/kg)	Total	Max. 0.5	0.0421	0	0.751	5	401	4	1.00%
	Inter-Andean		0.0310	0	0.751	4	343	1	0.29%
	Coast		0.0870	0.035	0.75	1	58	3	5.17%
Added Water (%)	Total	0	1.58	0	11.6	3	192	118	61.46%
	Inter-Andean		1.58	0	11.6	3	192	118	61.46%
Starch (µg/mL)	Total	0	0	0	0	4	204	0	0.00%
	Inter-Andean		0	0	0	4	204	0	0.00%
Antibiotics (µg/mL)	Total	LMR	-	-	-	21	2543	370	14.55%
	Inter-Andean		-	-	-	19	2447	356	14.54%
	Coast		-	-	-	2	96	14	14,58%
Arsenic (mg/L)	Total	0.01	0.00519	0	0.014	2	78	20	25.64%
	Inter-Andean		0.00003	0	0.001	1	58	0	0%
	Coast		0.01035	0.007	0.014	1	20	20	100%
Total Bacterial Count (CFU/mL)	Total	1.5 × 10^6^	6,878,541.1	130	296,000,000	33	109,610	22,338	20.38%
	Inter-Andean		7,522,151.9	130	296,000,000	29	109,270	22,229	20.34%
	Coast		5,713,750	680,000	43,700,000	2	136	67	49.26%
	Oriental		1,004,424.3	400	3,781,440	2	204	42	20.58%
Somatic Cell Count (SC/mL)	Total	7 x 10^5^	695,736.1	4000	14,330,000	30	110,347	878	0.80%
	Inter-Andean		766,050.3	4000	14,330,000	24	108,757	831	0.76%
	Coast		330,821.5	23261	1,723,895	4	1280	4	0.31%
	Oriental		572,898.7	18000	2,388,000	2	310	43	13.87%
Ash (%)	Total	Min. 0.65	0.6725	0.46	0.8	4	431	34	7.89%
	Inter-Andean		0.6650	0.46	0.8	2	386	34	8.81%
	Coast		0.68	0.65	0.71	2	45	0	0%
Chlorine (mg/mL)	Total	Max. 1500	0	0	0	3	657	0	0.00%
	Inter-Andean		0	0	0	3	657	0	0.00%
Colorants (µg/mL)	Total	0	0	0	0	1	132	0	0.00%
	Inter-Andean		0	0	0	1	132	0	0.00%
Cryoscopy (°C)	Total	−0.536 to –0.512	−0.535	−0.340	−0.760	8	1646	457	27.76%
	Inter-Andean		−0.760	−0.340	−0.534	7	1622	445	27.44%
	Coast		−0.5368	−0.513	−0.566	1	24	12	50%
Relative Density (g/mL)	Total	1.028–1.033	1.030	1.018	1.0372	22	1983	224	11.30%
	Inter-Andean		1.0313	1.026	1.033	18	1818	224	12.32%
	Coast		1.0313	1.026	1.033	4	165	0	0%
Eprinomectin (µg/kg)	Total	20	3.97	0	19.07	1	10	0	0.00%
	Inter-Andean		3.97	0	19.07	1	10	0	0.00%
Protein Stability	Total	Negative	-	-	-	10	1462	171	11.70%
	Inter-Andean		-	-	-	8	1433	169	11.79%
	Coast		-	-	-	2	29	2	6.90%
Glycomacropeptide (%)	Total	0	3.49	0.33	8.44	1	24	9	37.50%
	Inter-Andean		3.49	0.33	8.44	1	24	9	37.50%
Fat (%)	Total	Min. 3.0	3.69	0.26	10.33	35	109,428	564	0.52%
	Inter-Andean		3.74	0.26	10.33	30	109,195	563	0.52%
	Coast		3.35	1.9	5.35	4	165	1	0.61%
	Oriental		3.64	3.51	3.88	1	68	0	0%
Lactose (%)	Total	Min. 4.2	4.78	3.17	6.4	4	1097	56	5.10%
	Inter-Andean		4.78	3.17	6.4	4	1097	56	5.10%
Mercury (mg/L)	Total	0.006	0.00009	0	0.006	2	78	20	25.64%
	Inter-Andean		0.00009	0	0.002	1	58	0	0%
	Coast		0.0117	0.0018	0.006	1	20	20	100%
Neutralizants (µg/mL)	Total	0	-	-	-	7	951	137	14.41%
	Inter-Andean		-	-	-	7	951	137	14.41%
Peroxides (µg/mL)	Total	0	-	-	-	9	1011	61	6.03%
	Inter-Andean		-	-	-	8	987	61	6.18%
	Coast		-	-	-	1	24	0	0%
pH	Total	6.6–6.8	6.64	6.00	7.45	15	918	331	36.06%
	Inter-Andean		6.59	6.00	6.98	12	777	327	42.08%
	Coast		6.81	6.49	7.45	3	141	4	2.84%
Lead (mg/kg)	Total	Max. 0.02	0.208	0.16	0.719	1	58	57	98.28%
	Inter-Andean		0.208	0.16	0.719	1	58	57	98.28%
Protein (%)	Total	Min. 2.9	3.2	1.52	5	30	109,020	1012	0.93%
	Inter-Andean					25	108,787	1010	0.92%
	Coast		3.26	2.86	3.76	4	165	2	1.21%
	Oriental		3.29	3.27	3.31	1	68	0	0%
Ptaquiloside (mg/mL)	Total	0	0.648	0	2.64	3	128	114	89.06%
	Inter-Andean		0.648	0	2.64	3	128	114	89.06%
Reductase (hours)	Total	3	2.6	1.0	6	7	932	576	61.80%
	Inter-Andean		2.59	1.0	6	5	852	536	62.91%
	Coast		2.5	2.0	3.0	2	80	40	50.00%
Non-Fat Solids (%)	Total	Min. 8.2	8.66	4.03	10.56	13	1954	247	12.64%
	Inter-Andean		8.62	4.03	10.56	11	1862	247	13.27%
	Coast		8.88	8.45	9.28	1	24	0	0%
	Oriental		8.72	8.67	8.77	1	68	0	0%
Total Solids (%)	Total	Min. 11.2	12.24	4.61	20.1	22	106,707	511	0.48%
	Inter-Andean		12.49	4.61	20.1	17	106,402	503	0.47%
	Coast		11.26	7.77	13.74	4	165	8	4.85%
	Oriental		12.50	12.3	12.60	1	140	0	0%
Zearalenone (µg/L)	Total	30–1000	1.53	0	10.2	1	209	0	0.00%
	Inter-Andean		1.4	0.5	4.2	1	151	0	0.00%
	Coast		1.6	0	10.2	1	58	0	0.00%

**Table 2 foods-11-03351-t002:** Meta-analysis of variables, by sample and by region.

Variable	Meta-Analysis—Across Studies				Meta-Analysis—Region			
	Tau (τ)	I^2^	H^2^	*p*—Value of Q	Tau (τ)	I^2^	H^2^	*p*—Q Value Moderators
Titulable Acidity (°D)	2.58555	99.99%	10,011.64	<0.00001	0.0159	98.60%	71.25	0.09700
AFM1 (µg/kg)	0.00023	98.68%	75.61	<0.00001	0.00025	98.60%	71.25	0.097
Added Water (%)	117.515	99.37%	159.99	0.00136	Studies only conducted in the inter-andean region			
Starch (µg/mL)	0	0	0	0	0	0	0	0
Antibiotics (µg/mL)	0.01685	99.02%	102.50	<0.00001	0.01722	98.98%	98.28	0.38037
Arsenic (mg/L)	0.00005	99.81%	529.66	<0.00001	0.00005	99.81%	529.66	<0.00001
Total Bacterial Count (CFU/mL)	5.15	99.99%	12612.32	<0.00001	5.22	99.98%	5641.2	0.4445
Somatic Cell Count (SC/mL)	0.47	99.91%	1160.93	<0.00001	0.47	99.91%	1160.93	0.4228
Ash (%)	0.0444	99.10%	111	<0.00001	0.06327	99.84%	86.51	0.95386
Chlorine (mg/mL)	0	0	0	0	0	0	0	0
Colorants (µg/mL)	Not completed because there is only one study							
Cryoscopy (°C)	0.03461	99.90%	1018.06	<0.00001	0.03744	99.92%	1330.51	0.95819
Relative Density (g/mL)	0.00002	99.97%	3306.56	<0.00001	0.00794	99.96%	2605.54	0.59644
Eprinomectin (µg/kg)	Not completed because there is only one study							
Protein Stability	0.00385	90.17%	10.18	<0.00001	0.00437	91.67%	12.01	0.85649
Glycomacropeptide (%)	Not completed because there is only one study							
Fat (%)	0.56471	99.96%	2838.08	<0.00001	0.56404	99.96%	2696.38	0.31676
Lactose (%)	0.24647	99.55%	223.47	<0.00001	Studies only conducted in the inter-andean region			
Mercury (mg/L)	0.81976	99.76%	424.56	<0.00001	0.81976	99.76%	424.56	<0.00001
Neutralizants (µg/mL)	0.07833	99.70%	334.16	<0.00001	Studies only conducted in the inter-andean region			
Peroxides (µg/mL)	0.00228	93.02%	14.32	<0.00001	0.00286	94.86%	19.46	0.69234
pH	0.20532	99.81%	526.64	<0.00001	0.17826	99.72%	360.78	0.03261
Lead (mg/kg)	Not completed because there is only one study							
Protein (%)	0.13159	99.88%	849.11	<0.00001	0.134	99.81%	514.17	0.64311
Ptaquiloside (mg/mL)	0.33771	98.98%	97.80	<0.00001	Studies only conducted in the inter-andean region			
Reductase (hours)	0.81976	99.76%	424.56	<0.00001	0.71810	99.61%	254.08	0.18821
Non-Fat Solids (%)	0.38803	99.84%	638.05	<0.00001	0.40674	99.94%	93.98	0.67850
Total Solids (%)	1.27865	99.98%	5333.77	<0.00001	1.01577	99.96%	2581.76	0.00139
Zearalenone (µg/L)	Not completed because there is only one study							

## Data Availability

The data presented in this study are available on request from the corresponding author.

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
