# Peer review of "Milk Quality Parameters of Raw Milk in Ecuador between 2010 and 2020: A Systematic Literature Review and Meta-Analysis"

_foods, 2022, doi:10.3390/foods11213351_

Round 1
Reviewer 1 Report
foods-1898113
Article
Systematic literature review and meta-analysis of raw milk in Ecuador between 2010 and 2020
Authors have done a scientometric analysis / a systematic review and meta-analysis of 73 studies of raw milk produced in different regions of Ecuador between 2010 and 2020.
Very interesting
The objective of the research was to analyze the hygienic characteristics (bacteria, somatic cells, reductase), as well as the presence of contaminants (antibiotics, mycotoxins, heavy metals, preservatives, neutralizers) and adulterants (added water, starches, chlorides) in raw milk produced in Ecuador, through a
systematic literature review and meta-analysis, in studies conducted between 2010 and 2020.
Figures clarity si missing, I can understand as they were obtained from different software, but still to the possible extent they can be improved.
All is good with the manuscript, but some scientometric analysis is missing, authors may refere to some of the scientometric literature for possible further analysis and to include.
Author Response
Thank you very much for your recommendations. The corrections requested by all reviewers of the manuscript have been made. Regarding the figures, the request has been corrected. Regarding the scientometric analysis, we do not consider increasing it because the investigation is quite long and the fundamental thing has been answered with the statistics used. Corrected document is attached

Reviewer 2 Report
This article obtained milk parameters that meet standards and those that still need improvement by systematic review and meta-analysis of studies on raw milk parameters, which can help improve consumer health and profitability for producers. This article contains comprehensive literature, the workload is full, the analysis work is detailed and organized, the content of the chart is clearly expressed, and the research results are of social significance. The following are revision suggestions for reference.
1. The title cannot clearly indicate the content of the article. It is recommended to change it to: Milk quality parameters of raw milk in Ecuador between 2010 and 2020: A systematic literature review and meta-analysis.
2. The abstract presentation format can be presented according to the formation of purpose, method, result and conclusion.
3. Keywords:
(1) “parameters” is recommended to be modified to “milk quality parameters”.
(2) The keywords are suggested in order of importance: raw milk;milk quality parameters;Ecuador;meta-analysis.
4. Introduction:
(1) The introduction of the first paragraph on systematic reviews and meta-analysis is unnecessary and is recommended to be deleted.
(2) The content of the Introduction can be listed in the order of "Ecuadorian milk production, consumers' milk consumption, overall milk quality, the need to improve milk productivity in the context of globalization, current research on Ecuadorian milk, the purpose and content of this research".
5. Methods:
(1) The writing order is suggested to be changed to: search strategy (database, search deadline, search terms, search formula); literature inclusion criteria; literature quality/risk of bias evaluation (haven’t described in the text); literature data extraction; statistical methods.
(2) The process of study selection and searched results should be placed in the results section.
(3) All the searched documents that need to be analyzed do not need to be listed one by one in the text, but all the reference numbers need to be directly marked.
6. The writing order of Results and Discussion should be: the process of study selection and searched results; inclusion of literature characteristics/summary of the overall results of the literature; literature quality/risk of bias evaluation results (haven’t described in the text); meta-analysis results.
7. The references format is not completely standardized and not completely unified.
Author Response
Thank you very much for your recommendations. The corrections requested by all reviewers of the manuscript have been made. Corrected document is attached

Reviewer 3 Report
This research is focalized on a meta-analysis performed in Ecuador; The topic is neither relevant nor interesting to 98% of dairy scientists in the world.
The topic is original but not very interesting; the update to the international scientific literature is basically not relevant.
The paper is sufficiently well written and the text is easy to read.
The conclusions are consistent with the evidence and arguments presented, but these conclusions have only a local interest.
For all these reasons, I consider this manuscript not acceptable for an international journal ranking in the Q1 quartile. Considering the scientific importance of Ecuador as a possible case of study, I warmly suggest to the Authors to publish this meta-analysis study in a local journal, it will be surely better appreciated by the national scientists and by the local community academic society.
Author Response

(The authors gave the same response as above.)

Reviewer 4 Report
The authors examined a systematic meta-analysis of 73 studies of raw milk produced in different regions of Ecuador between 2010 and 2020. They reported the mean value of the physical-chemical characteristics of the milk (titratable acidity, ash, cryoscopy, fat, lactose, pH, protein, non-fat solids, and total solids) and also determined the great majority of these studies were within the range allowed by Ecuadorian regulations. The manuscript is well organized. The presented paper is interesting, but the following corrections should be done before publishing.
Below are my concerns and suggestions to improve the manuscript,
1. The abstract is missing a piece of brief information on the effects of contaminants and adulterants.
2. Introduction section is very poor This needs some editing.
3. In particular, aflatoxin M1 is a potent toxic mycotoxin that is classified as a Group 1 human carcinogen by the International Agency for Research on Cancer (IARC). Please add references with IARC.
4. Figures 6, 9, 11, 12, 14, and 17 should be arranged in some other way. It is not clear.
5. The conclusion needs to be highly quantitative and should be discussed in more detail.
6. I recommend adding the following current references related to the mycotoxin and meta-analysis in the introduction of this manuscript to improve its updated,
doi: 10.3934/agrfood.2022031
https://doi.org/10.4315/0362-028X.JFP-10-269
https://doi.org/10.3390/chemosensors9120363
Author Response
Thank you very much for your recommendations. The corrections requested by all reviewers of the manuscript have been made. The recommended bibliography has been used. Corrected document is attached

Reviewer 5 Report
Manuscript foods-1898113, entitled “Systematic literature review and meta-analysis of raw milk in Ecuador between 2010 and 2020”
This review article provides useful information about the characteristics of raw milk in Ecuador between 2010 and 2020. Some points should be corrected or clarified. There are also a lot of grammar, stylistic and syntax errors that in some cases negatively influence the understanding of the text.
General comments:
1. Please check the provided values for “AFM1”, “Total Bacterial Count”, “Glycomacropeptide”, “lead” etc. For example, for “AFM1” is it possible the min of measured level to be 0.0421, higher than the upper permitted limit by (NTE) INEN 9 and the percentage of samples not in compliance to be only 1%?
2. The forest plots by region are just a repetition of these by studies. Forest plots could be provided as supplementary material. Please provide a Table with means (min-max etc), as in Table 1, indicating the differences by region.
3. L101-130: Please provide these data as a supplementary material. For example, provide a Table with the following columns: name of authors, location, number of samples etc, as mentioned in L92-95.
4. Please check reference style (L223-224, 241, 256, 262-264, 271-274, 285-286, 292-293, 299-302, 316-316, 347, 357-358, 374, 377-378, 413-417, 422-425 etc)
Minor points
L14: “among” instead “between”
L16-17: “…had an influence…”
L17-18: “…greater in the Coastal compared to the Inter-Andean region.”
L18: “The mean values of…”
L20: “in” instead of “and”
L22: “…although the mean values were within these determined by local legislation…”
L23: “relatively higher” instead of “permissible”
L23-24: “…which possibly means that…”
L25: “…of several adulterants…”
L25: “confirmed” instead of “found”
L28-30: Please rephrase
L38-39: “…published in a variable…”
L40: “…types of reviews. In addition…”
L45: “…because the price of milk for the…”
L46: “necessity” instead of “aspiration”
L48-49: “…which makes it indispensable for the diagnosis of situations, the formulation and implementation of evaluation and…”
L55-56: “…period of time, so integration of information on a topic of interest has not been performed. Therefore, the general objective of the present research…”
L56: Please add “physico-chemical characteristics”
L63: Is it the word “population” appropriate? Maybe “Studies”.
L69: Please delete “of the same”
L81-82: Please delete “their equivalents in”
L89: “should” instead of “had to”
L134: Please delete “or”
L135: “…used. Statistical significance level was set on p <0.05.”
L136: “among” instead of “between”
L145: “should” instead of “must”
L147-148: Which is the difference?
Table 1 – 1st row: “Number of studies analyzed”, “Total samples”, “Number of samples out of range”, “Percentage of samples out of range”
L152: “As indicated by the analyzed studies, there is a…”
L157-158: “…above the maximum level permitted by Ecuadorian regulations. Similarly, according to the analysis for Mercury and Arsenic, it was found that 25.64% of samples (20/78) exceeded…”
L162: “them” instead of “these drugs”
L171: “was” instead of “were”
L175: “Among the 32…”
L180: “are intended” instead of “seek”
L189-190: “…lactose in 5.10% (56/1097), protein in 0.93% (1012/109 020), fat in…”
L199: “…rest of the parameters, it was observed that the heterogeneity index (I2), in all cases was higher…”
L200-202: Please rephrase
L205: “The results shown in Table 2” instead of “This leads us to”
L210: “among” instead of “between”
L214-215: “…and Total Solids, a p value of Q<0.05 was observed, indicating a higher presence in…”
L224-225: “…present values above, while the study of Villegas et al.[ ] below what is allowed.”
L231, 238: “variability” instead of “variance”
L232, 246: You mean “region”?
L236: “reported” instead of “have”
L238: Please delete “global”
L244: “Regarding the relative density…”
L247: Please delete “but”
L254: Figure 5C?
L255: “In the most studies regarding fat percentage, it was above the…”
L256: “However, there were studies, such these by Romero…”
L257-260: What do you mean? Please clarify.
L265: “were” instead of “are”
L272: “studies” instead of “research”
L320-321: “relatively higher” instead of “very permissible”
L326: “containing” instead of “having”
L337-338: “…its presence was higher than the minimum values allowed…”
L384: “carried out with the intention” instead of “done”
L476: “determined” instead of “described”
Author Response
Thank you very much for your recommendations. The corrections requested by all reviewers of the manuscript have been made.
AFM1 and lead values have been corrected. In the case of Glucomacropeptide and Total Bacterial Count, they are correct. The information for region has been included in Table 1. What is requested is delivered as complementary material. Reference style has been correted. In some cases, the author's surname (along with the year of publication) has been included, so that the discussion can be understood. Minor points has been correted.
Corrected document is attached

Round 2
Reviewer 3 Report
The citations in the text must be numbered, the Authors names must not be cited, only numbers in brackets.
Author Response
Thank you very much for your observations.
The requested correction has been made.

Reviewer 5 Report
Authors made some of the necessary amendments. However, some points were not corrected (comments in clean version):
1. The forest plots by region are just a repetition of these by studies.
2. Please check reference style (L247-455)
L32: Please delete "while"
L53: "for the producer"
L126-155: As stated in previous review round, this part is not necessary. Please make a reference to supplementary material
L164-165: Which is the difference? (previous review round)
Author Response

(The authors gave the same response as above.)
